# Regime-Switching Effect of Tourism Specialization on Economic Growth in Asia Pacific Countries

**Geng-Nan Chiang [1], Wei-Ying Sung [2],\* and Wen-Guu Lei [3]**

[1] Department of Finance, Feng Chia University, Taichung City 407, Taiwan; jnchiang@mail.fcu.edu.tw
[2] Institute of Public Affairs Management, National Sun Yat-sen University, Kaohsiung City 804, Taiwan
[3] Department of Tourism and Travel Management, Da-Yen University, Changhua County 515, Taiwan; greeley1234@yahoo.com.tw
\* Correspondence: jackerblack730112@gmail.com; Tel.: +886-912-153-997

**Abstract:** In the past 30 years, many studies have focused on exploring the relationship between tourism development and economic growth. However, there has been no consensus reached concerning of the relationship. This study will attempt to clarify the relationship between tourism development and economic growth. The purpose of this study is to analyze the relationship between tourism development and economic growth. This study applies the Panel Smooth Transition Regression Model (PSTR) proposed by Gonzalez et al. (2005) to investigate the regime-switching effect of tourism specialization on economic growth in Asia Pacific countries over the period 1996–2009. The results are as follows: (a) there were regime-switching effects of tourism specialization on economic growth; (b) the tourism specialization on economic growth has a better explanation for the effects of non-linear PSTR than linear PLS (Panel Least Squares); (c) in medium degree of tourism specialization countries (the value is between 0.0123~0.01663), tourism development has a significantly positive influence on economic growth, but consumption ability and investment ratios have a significantly negative influence on economic growth; (d) in low or high degree of tourism specialization countries (the value is below 0.0123 or above 0.01663), tourism development has a reduced influence on economic growth, and significantly positive influence on consumption ability and investment ratios. On the basis of these results, this study presents policy recommendations and areas for future research.

**Keywords:** tourism specialization; real international tourism receipts growth (% of GDP); economic growth; regime-switching effect; Asia Pacific countries; Panel Smooth Transition Regression Model (PSTR)

**JEL Classification:** R11

## 1. Introduction

### 1.1. Background and Motivation

Tourism has been one of the key factors influencing economic growth in most countries (Dwyer et al. 2004). According to surveys done by the World Tourism Organization, the number of global tourists reached 1.186 million and international tourism receipts amounted to 1260 billion US dollars in 2015 (World Tourism Organization 2016). Among all the regions in the world, the Asia Pacific countries have the fastest development of tourism. With the number of global tourists up to 279 million and international tourism receipts reaching 418 billion US dollars in 2015

(World Tourism Organization 2016), Asia Pacific countries have become the key indicators for global tourism development and economic growth.

Tourism development can facilitate the development of related industries and further boost the overall economic growth of a country (Lee and Chang 2008). Therefore, many countries take tourism development into account when making important policies on economic growth (Chou 2013; Chen and Chiou-Wei 2009; Oh 2005). In academia, more and more researchers have recently begun to examine the relationship between tourism development and the overall economic growth of a country (Chou 2013; Chen and Chiou-Wei 2009; De Vita and Kyaw 2016a; Dritsakis 2012; Lee and Chang 2008; Wang 2012a).

Previous studies indicate that the tourism development of a country has a direct influence on its economic growth (Dritsakis 2004; Lee and Chang 2008; Oh 2005; Yen 2012). An investigation of the relationship between tourism development and economic growth found that general economic factors, such as price level and investment ability, have an indirect effect on the economic growth of a country (Po and Huang 2008). Regarding tourism development and economic growth, Lee and Chang (2008) suggest that the international tourism receipts of a country have a significant influence on its GDP growth. Chao et al. (2006) suggest that, while the promotion of tourism development might bring substantial international tourism receipts to a country, certain crowding-out effects may take place in its economy.

Several studies show that the relationship between tourism development and economic growth have different effects on countries with a range of economic developments (Bilen et al. 2017; De Vita and Kyaw 2016a; Eugenio-Martin et al. 2004; Holzner 2011; Lee and Chang 2008; Wang 2012b), i.e., tourism development does not necessarily have a positive influence on the economic growth of a country; in fact, it may even have a negative impact, to certain degree. The situation may depend on the economic development and tourism specialization of a particular country (Chou 2013; De Vita and Kyaw 2016b; Sequeire and Campo 2005; Brau et al. 2007; Po and Huang 2008).

Lee and Chien (2008) indicate that there has been no consensus regarding whether tourism development has a positive or negative effect on the economic growth of a country. Through a literature review, this study found that most studies have used a linear model to explore the relationship between tourism development and economic growth; few studies have adopted a non-linear model for their examination. However, the linear model fails to eliminate such problems as short-term economic fluctuations and structural changes, when used to investigate the relationships among variables (Po and Huang 2008). The non-linear model should be applied not only to eliminate the problems, but also to better understand the effects between two variables. Therefore, this study used the Panel Smooth Transition Regression Model (PSTR) to investigate the regime-switching effect of tourism specialization on economic growth in Asia Pacific countries, where tourism has developed the fastest rates in recent years. In sum, through both a linear and a non linear model analysis, this study explored the relationship between tourism development and economic growth among different stages of tourism specialization. Moreover, this study compared the prediction effects of the traditional linear model and non-linear model to present policy recommendations and potential areas for future research.

*1.2. Purpose*

Based on the aforementioned research background and motivations, the research purposes are as follows.

1. To understand the regime-switching effect of tourism specialization in Asia Pacific countries on their economic growth.

2. To compare the prediction effects of the linear model and the non-linear model in regard to the influence of the tourism development in Asia Pacific countries on their economic growth.

## 2. Literature Review

In the past 30 years, many studies have focused on exploring the relationship between tourism development and economic growth (Chou 2013) and various hypotheses have been proposed. Some have suggested that tourism development would influence economic growth (Wang 2012b), some have indicated that economic growth would affect tourism development (Narayan 2004), and still others have suggested that tourism development and economic growth affect each other (Bilen et al. 2017; Chen and Chiou-Wei 2009; Lee and Chang 2008). However, there is no real consensus regarding the relationship between tourism development and economic growth (Oh 2005). Reviewing the previous studies, the authors of this study found that the relationship between tourism development and economic growth can be divided into three categories, as follows.

The first category of studies is devoted to the exploration of causality in the relationship between tourism development and economic growth of a single country. Balaguer and Cantavella-Jorda (2002) used Granger's causality test to investigate the relationship between tourism development and economic growth in Spain, and discovered that tourism development has a positive influence on economic growth. Dritsakis (2004) explored the relationship between tourism development and economic growth in Greece and found that they have mutual influence, with the actual exchange rate and the international tourism receipts having the most significant effects on economic growth. Durbarry (2004) analyzed the influence of tourism development in Mauritius on the country's economic growth, through the gross domestic fixed capital formation, trade exports and real international tourism receipts. The results showed that all three factors had a significantly positive influence on the country's economic growth. Narayan (2004) analyzed the influence of tourism development in Fiji on its economic growth and found that tourism development leads to the appreciation of the local currency's exchange rate and rising prices of goods. Kim et al. (2006) analyzed the relationship of tourism development and economic growth in Taiwan; the results showed that economic scale and trade openness affect economic growth.

Most studies in the first category adopted Granger's causality test to examine the relationship between tourism development and economic growth; however, this method may lead to biased estimates because of insufficient sample data, short-term economic fluctuations, and the inability to show features of different countries. To solve this problem, some researchers have started to apply panel data in examining the relationship between tourism development and economic growth among various countries, which forms the second category of studies. For example, Bilen et al. (2017) analyzed the relationship between tourism development and economic growth in Mediterranean countries. The results showed a bidirectional causality between tourism and economic growth. Eugenio-Martin et al. (2004) analyzed the relationship between tourism development and economic growth in Latin American countries and found that tourism development has a positive influence on economic growth in low and middle income countries, but no effect on economic growth in high income countries. However, when Lanza et al. (2003) analyzed the relationship of tourism development and economic growth in 13 OECD countries, they discovered that tourism development positively influenced economic growth in developed countries. Lee and Chang (2008) further compared the influence of tourism development on OECD countries and non-OECD countries. Their results showed that international tourism receipts have a greater influence on the GDP of non-OECD countries than that of OECD countries, and that the currency exchange rate significantly affects the economic growth in both OECD and non-OECD countries. Yen (2010) investigated the top nine most-visited countries and found that tourism development did not influence economic growth. Wang (2012b) used the threshold effect to examine the relationship between tourism development and economic growth in 10 countries (growth rate of international tourism receipts as threshold variable). The results showed that currency exchange rates have a positive influence on the economic growth in high-threshold countries, and that inflation suppresses economic growth in both high-threshold and low-threshold countries.

From the second category of studies, it is clear that although researchers tried to solve the problem of insufficient sample data by adding panel data, short-term economic fluctuations and structural changes could not be eliminated (Po and Huang 2008).

As a result, some researchers started trying the non-linear model to solve the aforementioned problem, which forms the third category of studies. For example, Po and Huang (2008) adopted the threshold vector autoregressive model in which tourism specialization was used as the threshold variable to analyze the relationship between tourism development and economic growth in 88 countries. The results suggested that those 88 countries could be divided into three regimes. In Regime 1 and Regime 3 (lower than the low threshold and higher than the high threshold), tourism development had a positive influence on economic growth. In Regime 2 (within the low threshold and high threshold), although tourism development did not have a significant influence, further analysis showed that tourism development still had a positive influence on economic growth. Chang et al. (2010) adopted the Panel Threshold Regression Model in which tourism specialization was used as the threshold variable to examine the relationship between tourism specialization and economic growth in 131 countries. The results showed that those 131 countries could be divided into three regimes. Among them, tourism development would have a significantly positive influence on economic growth in low-regime and middle-regime countries, while it would not significantly impact the economic growth in high-regime countries. Yen (2012) also adopted the Panel Threshold Regression Model to analyze the relationship between tourism development and economic growth in 84 countries. The results showed that those 84 countries could be grouped into two categories: high-threshold countries and low-threshold countries. Tourism development had a positive influence on economic growth in both high-threshold countries and low-threshold countries, while trade openness had a negative influence on their economic growth. De Vita and Kyaw (2016a) used the system generalized methods-of-moments (SYS-GMM) estimation methodology to investigate the tourism-growth relationship for a large panel of 129 countries. The results showed that they could be divided into three categories: low-income countries, middle-income countries and high-income countries. Among them, tourism development had a significantly positive influence on economic growth in low-income countries, middle-income countries and high-income countries.

From the third category of studies, it can be concluded that the non-linear model can solve the problem of biased estimates by eliminating insufficient sample data, short-term economic fluctuations and structural changes. The authors of this paper reviewed the previous studies and found that few studies applied the non-linear model to explore the relationship between tourism development and economic growth. This may be one of reasons why the previous studies failed to clearly define the relationship between tourism development and economic growth. As the result, this study aims to adopt tourism specialization as the threshold variable, and use cross-sectional data and the Panel Smooth Transition Regression Model (PSTR) to examine the relationship between tourism development and economic growth. Moreover, this study will compare the differences of prediction effects of both the traditional linear model, and the non-linear model.

In the past, there has been no consensus reached concerning the relationship between tourism development and economic growth, which may be related to the nonlinear relationship between them. Therefore, this study has adopted the non-liner model to examine the relationship between tourism development and economic growth. This study further clarifies the influence of different stages of tourism development and economic growth. On the basis of the results found in this study, the authors provide policy recommendations and areas for future research.

## 3. Methodology

### 3.1. Data Sources

The research data in this study came from the World Development Indicators database (WDI) from the World Bank (2014); the International Financial Statistics (IFS) from the International Monetary Fund (2014);

the Economic Data (ED) from the World Travel & Tourism Council (WTTC) and the United Nations (2014). To maintain the consistency and completeness of the data in this study, the researcher eliminated countries and time periods with missing values, and selected 33 Asia Pacific countries over the period 1996–2009 as data for the research sample.

*3.2. Definition and Measurement of Variables*

3.2.1. Threshold Variables

Tourism specialization is used as the threshold variable in this study; it is defined as the percentage of international tourism receipts in GDP. From the literature review, it was found that tourism specialization has often been used as the threshold variable. For example, Sequeire and Campo (2005), Brau et al. (2007), Po and Huang (2008), Chang et al. (2010), Yen (2012) and Kung (2013) all used tourism specialization as the threshold variable in their studies, despite having different definitions. Sequeire and Campo (2005) used "the percentage of tourism receipts in GDP", "the percentage of tourism receipts in the export of goods and labor" and "the percentage of global tourists in the total population" to measure the tourism specialization of a country. Chang et al. (2010) adopted "the percentage of real tourism GDP in real GDP" to measure the tourism specialization of a country. Brau et al. (2007), Po and Huang (2008) and Yen (2012) defined tourism specialization as "the percentage of international tourism receipt in GDP". Considering that most previous studies defined tourism specialization as the percentage of international tourism receipts in GDP, and such a definition is more suitable for this study's examination of economic growth of countries in a specific region, this study adopted the definition used in Brau et al. (2007), Po and Huang (2008) and Yen (2012).

3.2.2. Explanatory Variables

This study used the growth rate of international tourism receipts (TRG), the percentage of the gross fixed capital formation in GDP (I) and inflation rate ($\pi$) as explanatory variables (see Table 1). It was found through the literature review that the aforementioned variables have often been used as explanatory variables of economic growth (Kung 2013; Yen 2012). For example, Yen (2012) adopted "the growth rate of international tourism receipts", "the percentage of the gross fixed capital formation in GDP" and the "inflation rate" as explanatory variables to predict economic growth, when exploring the relationship between tourism development and economic growth of various countries.

**Table 1.** Definitions of explanatory variables.

| Variables | Definitions | Remarks |
|---|---|---|
| EG | economic growth | "The growth rate of the GDP per capita" from the World Development Indicators (WDI) of the World Bank was used to measure the economic growth of a country in this study. |
| TRG | growth rate of international tourism receipts | "The real international tourism receipts" was used to measure the tourism development of a country in this study. |
| $\pi$ | inflation rate | "Inflation, GDP deflator" from the WDI of the World Bank was used to measure the price level of a country in this study. |
| I | the percentage of the gross fixed capital formation in GDP | "The percentage of the gross fixed capital formation in GDP" from the WDI of the World Bank was used as the proxy variable of the real capital investment in this study. |
| q | degree of tourism specialization | "The percentage of international tourism receipts in GDP" was used to measure the tourism specialization of a country in this study. |

**Data source:** The World Development Indicators database (WDI) of the World Bank (2014).

### 3.3. Model Estimation and Tests

#### 3.3.1. Panel Data Least Squares Regression (PLS)

The study applies the Panel Data Least Squares Regression (PLS) by Hsiao (1986) proposed to investigate the effect of tourism specialization on economic growth in a general linear model. In order to reduce the heterogeneity bias of a single sample, and control the autocorrelation between time serious and individual differences in panel data, the study used the fixed effect of PLS to analyze panel data as follows:

$$y_{it} = \alpha_i + \sum_{k=1}^{k} \beta_k X_{kit} + \varepsilon_{it} \tag{1}$$

where $i$ represents different countries, $i = 1 \ldots N$; $t$ is the observation time state, $t = 1 \ldots T$; $k$ is the number of countries, $k = 1 \ldots k$; $y$ is the vector; **B** is the regression coefficients of explanatory variables; $\chi$ is the vector of explanatory variables; $\varepsilon_{it}$ error item; $\alpha_i$: intercept, individual effect, not change with time, which had different effect in other units.

#### 3.3.2. Panel Smooth Transition Regression (PSTR) Model

This research intends to understand the regime-switching effect of tourism specialization on economic growth if had a smooth transition threshold effect. This study constructs the PSTR model by Gonzalez et al. (2005), and can be defined as:

$$y_{it} = \mu_i + \beta_0' X_{it} + \beta_1' X_{it} g(q_{it}; \gamma, c) + \varepsilon_{it} \tag{2}$$

where $i = 1, \ldots, N$, $t = 1, \ldots, T$, and $N$ and $T$ stand for the cross-section and time dimensions of the panel, respectively. The dependent variable $y_{it}$ is a scalar; $i$ represents the fixed individual effect; $X_{it}$ is a k-dimensional vector of time-varying exogenous variables; $\mu_{it}$ is the residual term. The transition function $g(q_{it}; \gamma, c)$ is a continuous function of the observable variable $q_{it}$. It is normalized to be bounded between 0 and 1; these extreme values are associated with regression coefficients $\beta_0'$ and $\beta_0' + \beta_1'$. The value of $q_{it}$ determines the value of $g(q_{it}; \gamma, c)$ and thus the effective regression coefficients $\beta_0' + \beta_1'$. $g(q_{it}; \gamma, c)$ for any individual $i$ at time $t$.

Following Granger and Teräsvirta (1999), Teräsvirta (1994), and Jansen and Teräsvirta (1996), we have formulated the transition function as follows:

$$g(q_{it}; \gamma, c) = \left\{ 1 + exp \left[ -\gamma \prod_{j=1}^{m} (q_{it} - c_j) \right] \right\}^{-1} \quad with \ \gamma > 0 \ and \ c_1 \leq c_2 \leq \ldots \leq c_m \tag{3}$$

where c $(c_1, \ldots, c_m)$ is an m-dimensional vector of location parameters and the slope parameter determines the smoothness of the transitions. In general, it is sufficient to consider $m = 1$ or $m = 2$, as these values allow for commonly encountered types of variations in the parameters. In the case of $m = 1$, the model specifies that the two extreme regimes are associated with low and high values of $q_{it}$ with a single monotonic transition of the coefficients from $\beta_0'$ to $\beta_0' + \beta_1'$ as $q_{it}$ increases, such that the change is centered around $c_1$. In the case of $m = 2$, the transition function has its minimum at $(c_1 + c_2)/2$ and reaches the value 1 at both low and high values of $q_{it}$. When this approaches infinity, the PSTR model reduces to a three-regime panel threshold regression (PTR) model with identical outer regimes and a different middle regime (Gonzalez et al. 2005).

The multi-level PSTR model is a generalization of the PSTR model that allows for more than two different regimes; it can be formulated as:

$$y_{it} = \mu_i + \beta_0' X_{it} + \sum_{j=1}^{r} \beta_0' X_{it} g_j \left( q_{it}^j; \gamma_j; c_j \right) + \varepsilon_{it} \tag{4}$$

where the transition functions $g_j(q_{it}; \gamma, c), j = 1, ..., \gamma$ depend on the slope parameters $\gamma_j$ and on location parameters $C_j$. If $r = 1$, $q_{it}^j = q_{it}$, and $\gamma_j \to \infty$ for all $j = 1, ..., \gamma$ then the transition function becomes an indicator function, with I[A] = 1 when event A occurs, and I[A] = 0 otherwise; in such a case, the model in Equation (4) becomes a PTR model with r + 1 regimes. As a result, the multi-level PSTR model can be viewed as a generalization of the multiple regime panel threshold model (PTR) in Hansen (1999).

### 3.3.3. Building the Panel Smooth Transition Regression Model

The PSTR model building procedure consists of specification, estimation and evaluation stages. Specification includes tests for homogeneity, and selection of the transition variable $q_{it}$. If the tests fail to show homogeneity, then specification includes the determination of the appropriate form of the transition function; the form is dictated by the value of m in Equation (3). A nonlinear least square method is used for parameter estimation. At the evaluation stage the estimated model is subjected to misspecification tests to check whether it provides an adequate description of the data. The null hypotheses to be tested at this stage includes parameter constancy, absence of remaining heterogeneity and absence of autocorrelation in the errors. Finally, the number of regimes in the panel must be specified, which means that a value must be assigned to r in Equation (4).

## 4. Data Analysis

### 4.1. Basic Descriptive Statistical Analysis of the Variables

It is clear from Table 2, that the average EG of the 33 Asia Pacific countries is 3.87%, the average TGR is 18.74%, the average $\pi$ is 9.6%, the average I is 23.53% and the average q is 3.68%. Except for the obviously higher number of the average TGR, the results in this study are similar to those of Yen (2012)'s analysis of the relationship between tourism development and economic growth of all countries in the world. It is suggested that such a difference may arise from the different sampling sizes. Yen (2012) adopted 84 countries in the world as the research sample data while this study only targeted Asia Pacific countries. Since the Asia Pacific countries have had such rapid tourism development over recent years, the international tourism receipts in this region are higher than those in most places of the world (Europe was ranked first, Asia Pacific was ranked second) (World Tourism Organization 2016).

**Table 2.** Basic descriptive statistical analysis.

| Variables | Mean | Std. Dev | Variance | Max | Min | Kurtosis | Skewness |
|-----------|------|----------|----------|-----|-----|----------|----------|
| EG | 3.8697 | 4.6711 | 1.2071 | 33.0305 | −14.3851 | 5.5259 | 0.3277 |
| TRG | 18.7434 | 92.5802 | 4.9394 | 1820 | −50.8772 | 309.55 | 16.293 |
| $\pi$ | 9.6028 | 15.3534 | 1.5988 | 137.9649 | −21.4438 | 22.328 | 3.8647 |
| I | 23.5348 | 11.4895 | 0.4882 | 63.0487 | −89.8562 | 30.949 | −3.3547 |
| q | 3.6767 | 4.67 | 1.2702 | 35.167 | 0.0879 | 11.498 | 2.9909 |

### 4.2. Correlation Analysis of the Variables

From Table 3, it is obvious that the correlation coefficient of all explanatory variables falls between −0.2800 and 0.1172; the correlation coefficient of EG and TRG is 0.0449; the correlation coefficient of EG and $\pi$ is −0.0136; the correlation coefficient of EG and I is 0.1172. The results indicate that there is a significantly positive correlation between EG and I ($r = 0.1172$, $p < 0.05$) at the 10% confidence level, while there is no significant correlation between EG and other explanatory variables. It can be concluded that despite the low correlation between economic growth and the explanatory variables in the study, the correlation is similar to that of previous studies (Yen 2012; Kung 2013).

**Table 3.** Variables correlation analysis.

|      | EG       | TRG      | Π          | I   |
|------|----------|----------|------------|-----|
| EG   | 1        |          |            |     |
| TRG  | 0.0449   | 1        |            |     |
| Π    | −0.0136  | −0.0150  | 1          |     |
| I    | 0.1172 * | −0.0187  | −0.2800 ** | 1   |

Note: *, ** and *** denote significance at the 10%, 5% and 1% level, respectively.

### 4.3. Analysis of Panel Unit Root Test

To enhance the accuracy of the results, data used in an econometric model should be confirmed to be stationary before the model is established and estimated through a time series (Nelson and Plosser 1982). The structure of the sample data used in this study is balanced panel data. Therefore, a panel unit root test of the research data should be done before data analysis.

This study adopted the most used Fisher-type augmented Dickey–Fuller test (ADF) to test the stationary state of the variables in this study (Maddala and Wu 1999). Moreover, because the ADF unit root test does not take into consideration the autocorrelation and ARCH/GARCH of residuals, this study also adopted the PP–Fisher test, proposed by Phillips and Perron (1988), to enhance the test results.

From Table 4, the results from the three tests all reject the null hypothesis of the unit root, i.e., the sample data in this study are stationary. Thus, an analysis of the linear regression model and the panel smooth transition regression model could be conducted.

**Table 4.** Panel unit root test.

|     | Augmented Dickey–Fuller (ADF)–Fisher Test (*p*-Value) | PP–Fisher Test (*p*-Value) |
|-----|------------------------------------------------------|----------------------------|
| EG  | 135.6643 *** (0.0000)                                | 120.6493 *** (0.0000)      |
| TRG | 136.2843 *** (0.0000)                                | 221.1713 *** (0.0000)      |
| π   | 115.9583 *** (0.0000)                                | 185.2743 *** (0.0000)      |
| I   | 92.18593 *** (0.0005)                                | 75.18653 ** (0.00194)      |

Note: *, ** and *** denote significance at the 10%, 5% and 1% level, respectively.

### 4.4. Analysis of the Ordinary Least-Squares (OLS) Regression for Panel Data

From the test results in Table 5, it is clear that only the influence of the gross fixed capital formation in GDP ($\beta$ = 0.0505, $p < 0.05$) on economic growth reached a significant level. TGR, the most important representative of tourism development, did not achieve a significant influence on economic growth ($\beta$ = 0.0024, $p > 0.05$), thereby showing that tourism development does not necessarily have a direct impact on economic growth. This result differs from the conclusion in most previous studies that tourism development has a direct influence on economic growth (Balaguer and Cantavella-Jorda 2002; Dritsakis 2004; Kung 2013; Lee and Chang 2008; Oh 2005), but is similar to Yen (2010)'s conclusion that tourism development does not have a significant influence on economic growth. This result also presents the reason why the previous studies failed to clearly define the relationship between tourism development and economic growth. It suggests that the linear model may not fully explain the relationship between tourism development and economic growth. It may take a non-linear model to further understand the relationship between tourism development and economic growth.

**Table 5.** Linear regression model test (adjusted $R^2$ = 0.0164, *F*-statistic (*p*-value) = 2.54 (0.63)).

| Explanatory Variables | $\beta_0$ (*t*-value) |
|:---:|:---:|
| TRG | 0.0024 (1.03) |
| $\pi$ | 0.0067 (0.45) |
| I | 0.0505 ** (2.57) |
| CONS | 2.5723 *** (4.55) |

Note: ** and *** denote significance at the 5% and 1% level, respectively.

### 4.5. Analysis of the Panel Smooth Transition Regression Model (PSTR)

This study further used the PSTR to test the influence of tourism development on economic growth. In this study, tourism specialization was used as the transformation variable to analyze the relationship between tourism development and economic growth. First, the test for homogeneity was conducted to check the linear test of Asia Pacific countries. From the results of the test for homogeneity in Table 6, it can be noted that the effect of tourism specialization in Asia Pacific countries on economic growth rejected the null hypothesis of the linear model (LRT = 10.968, *p* < 0.001), so a non-linear model was accepted, i.e., the model in which tourism specialization is used as the transform variable should be a non-linear model.

Furthermore, a proper PSTR transform model should be selected for this study. From the comparison of results for when *m* = 1 and when *m* = 2 in Table 6, it is clear that BIC when *m* = 1 (BIC = 2.8271) is smaller than that when *m* = 2 (BIC = 2.8294) but both RSS and AIC when *m* = 2 (RSS = 6792; AIC = 2.7488) are smaller than those when *m* = 1 (RSS = 6882; AIC = 2.7554). Because there are three explanatory variables in the model, it is better to adopt a model where AIC is smaller (Gonzalez et al. 2005). Moreover, the linear null hypothesis cannot be rejected when *m* = 1 (LRT = 0.723, *p* > 0.05), so the model should be the model of Exponential PSTR, *m* = 2, when tourism specialization is used as the transform variable.

**Table 6.** Tourism specialization nonlinear model selection.

| Threshold Number | *m* = 1 | *m* = 2 |
|:---:|:---:|:---:|
| RSS | 6882 | 6792 |
| AIC | 2.7554 | 2.7488 |
| BIC | 2.8271 | 2.8294 |

Note: RSS denotes Residual square sum; AIC denotes Akaike information criterion; BIC denotes Bayesian information criterion.

Note:RSS= Residual square sum AIC=Akaike information criterion BIC=Schwartz's Bayesian information criterionThis study further examined the number of the conversion ranges in the test model. From the results in Table 7, when tourism specialization is used as the transform variable, the *m* = 2 model does not reject the *r* = 1 null hypothesis. Therefore, when tourism specialization is used as the transform variable, the PSTR model should be set up as *m* = 2, *r* =1.

**Table 7.** Tourism specialization nonlinear model range number.

| H$_0$: PSTR with *r* = 1 | |
|:---:|:---:|
| **Threshold Number** | ***m* = 2** |
| Wald Test | 1.653(0.949) |
| Fisher Test | 0.250(0.959) |
| LRT Test | 1.656(0.949) |

In the end, after the test of the PSTR model in which tourism development is used as the transform variable, it can be noted from the results in Table 8 that there are two thresholds in this model: 0.1663 and 0.0123, i.e., 0.1663 and 0.0123 are the threshold values of tourism specialization of the countries in this study. If tourism development of a country is lower than 0.0123 ($m < 0.0123$), the country is in the low tourism specialization; when it is higher than 0.1663 ($m > 0.1663$), the country is in the high tourism specialization; when it is between 0.0123 and 0.1663, the country is of the intermediate level of tourism specialization. When tourism specialization is between 0.0123 and 0.1663, TRG ($\beta = 0.0128$, $p < 0.01$), $\pi$ ($\beta = -0.0431$, $p < 0.01$) and I ($\beta = -0.073$, $p < 0.01$) reach the level of significance. Moreover, TRG positively influences economic growth, while $\pi$ and I negatively influences economic growth, i.e., in those Asia Pacific countries of intermediate tourism specialization, tourism development has a positive influence on economic growth, while the price level and investment proportion exerts a negative influence on economic growth. This result is similar to the conclusion of Yen (2012) and Po and Huang (2008) that tourism development has a positive effect on the economic growth of countries within the two threshold values of tourism development.

**Table 8.** Tourism specialization nonlinear regression model test.

| Threshold Number | $C_1 = 0.1663$ | Transform Speed | $\gamma = 2.8444 \times 10^3$ |
|---|---|---|---|
| | $C_2 = 0.0123$ | | |
| Transform Range | $0.0123 \leq m \leq 0.1663$ | $m < 0.0123$ or $m > 0.1663$ | |
| Explanatory Variables | $\beta_0$ (*t*-value) | $\beta_1$ (*t*-value) | $\beta_0 + \beta_1$ |
| TRG | 0.0128 ** (2.0490) | −0.0146 ** (−1.9427) | 0.0028 ** |
| Π | −0.0431 *** (−2.9650) | 0.0943 *** (3.1008) | 0.0512 *** |
| I | −0.0730 *** (−2.3829) | 0.1115 *** (3.2213) | 0.0385 *** |

Note: *, ** and *** denote significance at the 10%, 5% and 1% level, respectively. $C_1$ & $C_2$ = Threshold Number; $\gamma$ = Transform Speed. TRG = Growth Rate of International Tourism Receipts; $\pi$ = Inflation Rate; I = The Gross Fixed Capital Formation/GDP.

It can be noted from Table 8 that in the Asia Pacific countries of low tourism specialization and high tourism specialization, TRG ($\beta = -0.0146$, $p < 0.05$), $\pi$ ($\beta = 0.0943$, $p < 0.01$) and I ($\beta = 0.1115$, $p < 0.01$) also reach the significant level. However, TRG's original positive influence on economic growth is lowered, while $\pi$'s and I's original negative influence on economic growth becomes positive. In other words, in those Asia Pacific countries of low tourism specialization and high tourism specialization, the positive influence of tourism development on economic growth is reduced while price level and capital investment proportion changes from negative to positive. In tandem with the research results of Po and Huang (2008), it is suggested that tourism development, price level and capital investment proportion all have a positive influence on economic growth in countries whose the threshold value is lower than the low threshold of tourism specialization, and in countries whose threshold value is higher than the high threshold of tourism specialization.

From the above analysis results, tourism specialization has a regime-switching effect on the economic growth in the Asia Pacific countries. Moreover, in comparison with the Ordinary Least-Squares (OLS) regression for panel data (see Table 5), the non-linear PSTR model has better explanatory effect than the linear regression model. It can be further inferred that while the frequently-used linear model did not fail to define the relationship between tourism development and economic growth, it did tend to determine the relationship between tourism development and economic growth in a general way. Accordingly, it could be seen that tourism development would influence economic growth (Lanza et al. 2003; Eugenio-Martin et al. 2004; Wang 2012b), that tourism

development would not influence economic growth (Yen 2010) and that there would be mutual influence on each other (Chen and Chiou-Wei 2009; Holzner 2011). However, this study shows that in the process of tourism development and economic growth of a country, the original positive effect of some variables will transform into negative effects and vice versa. These two effects will reduce and enhance each other mutually. Owing to the limitation of the methodology (failure to eliminate short-term economic fluctuations and structural changes), the linear regression model failed to explain the reasons behind the results. This result is similar to some researchers' suggestion that a non-linear model should be employed to eliminate the limitations (Po and Huang 2008; Yen 2012).

In addition, this study found that the price value and capital investment proportion have a negative influence on economic growth in countries within the two threshold values of tourism specialization, while they have a positive influence on economic growth in countries whose threshold values are lower than the low threshold of tourism specialization, and in countries whose threshold values are higher than the high threshold of tourism specialization. This result may be due to the fact that inflation and capital investment proportion have different impacts in countries of various economic growth levels. From the perspective of the general economy, a moderate inflation rate and investment proportion will facilitate economic growth, but this is not the case in conditions of excessive inflation and investment proportion. In this study, most of the countries within the threshold values of tourism specialization are developing countries. Excessive inflation and investment proportion are prone to impact their developing economy with a negative effect on their economic growth. Most of the countries of low tourism specialization and countries of high tourism specialization in this study are underdeveloped countries. With a lack of infrastructure, they need certain capital investment and tourism receipts. A large amount of capital investment and a great number of tourists leads to inflation. Accordingly, inflation and capital investment proportion have different impacts on countries of different tourism development levels.

## 5. Conclusions and Suggestions

According to the analysis in this study, conclusions and suggestions are as follows:

### 5.1. Conclusions

A.  There were regime-switching effects of tourism specialization on economic growth in Asia Pacific countries.
B.  The non-linear PSTR has better explanatory effects than the linear PLS regarding the influence of tourism development on economic growth of Asia Pacific countries. In general, the relationship between tourism development and economic growth can be more clearly understood by using a non-linear model than linear models.

In Asia Pacific countries, where their threshold values of tourism specialization fall between 0.0123 and 0.1663, tourism development has a significantly positive influence on economic growth, while price level and investment proportion have a negative impact on economic growth. That is to say, in those Asia Pacific countries of intermediate tourism specialization, tourism development still promotes economic growth, but it may impact on export trade and foreign investment intentions.

In Asia Pacific countries where the threshold values of tourism specialization are lower than 0.0123 or higher than 0.1663, the original positive influence of tourism development on economic growth is reduced, while the original negative effect of price level and investment proportion becomes positive concerning economic growth.

### 5.2. Suggestions on Policy

Based on the aforementioned research results, it is clear that tourism development is not always suitable for all countries regarding their economic growth. The best development strategy is to enact

appropriate economic policies depending on the tourism specialization of a country. Therefore, based on the analysis results, this study presents the following related suggestions on policy-making.

1.   Asia Pacific countries of intermediate tourism specialization can adopt aggressive tourism policies while lowering the price level and investment proportion to facilitate economic growth.
2.   Asia Pacific countries of low tourism specialization and Asia Pacific countries of high tourism specialization can adopt conservative tourism policies while increasing the price level and investment proportion to facilitate economic growth.

*5.3. Suggestions for Follow-Up Research*

First, listing Per Capita GDP (PPP) as a threshold variable in investigation of the relationship between tourism development and economic growth.

From previous studies, it was found that most researchers listed PPP as an explanatory variable, while only Yen (2012)adopted PPP as the threshold variable to group countries into rich countries and poor countries in the investigation of the relationship between tourism development and economic growth. Few studies did further research on this aspect. From the related previous studies, it is clear that PPP is one of the key variables in tourism behavior (Holzner 2011). Therefore, it is suggested that follow-up studies can adopt PPP as a threshold variable to explore its influence on the relationship between tourism development and economic growth. Moreover, it is suggested that PPP can be used as a criterion to classify countries of different income, and to compare different classifications of these countries. In Yen (2012)'s study , the threshold value of PPP is 2674 dollars. Countries where the PPP threshold value is higher than 2674 dollars are classified as rich countries, while countries where the PPP threshold value is lower than 2674 dollars are classified as poor countries. This classification greatly differs from the World Bank's classification of countries of different income. Thus, it is suggested that threshold values generated by PPP can be used in the future to compare with the current criterion to enhance the objectivity of the classification of countries of different income.

Second, examining other variables in the investigation of the relationship between tourism development and economic growth.

The explanatory variables in the study are the growth rate of the international tourism receipts (TRG), the gross fixed capital formation in GDP (I) and the inflation rate ($\pi$). However, from previous studies, it was found that many other variables were used as explanatory variables (such as currency exchange rate, trade amenability, financial development, employment rate, and so on). Therefore, it is suggested that other variables be used in future studies to investigate their influence on economic growth in the field of tourism specialization.

Third, exploring the regime-switching effect of tourism specialization on economic growth in different regions.

This study aimed to explore the regime-switching effect of tourism specialization on economic growth across Asia Pacific countries, while most previous studies focused on a single country, single union country (like OECD), and even all countries in the world (see Table 1). Few studies have made comparisons of different regions. In light of this, it is suggested that the regime-switching effect of tourism specialization on economic growth in different regions can be explored in the future to find the best level of tourism specialization for different regions, and to present appropriate policies.

Fourth, following up time series data in investigations of the relationship between tourism development and economic growth.

In order to maintain the consistency and completeness of the data in this study, the authors adopted balance panel data over the period 1996–2009 as the research sample data. This was because parts of the sample data had missing values in the time period. This is considered a limitation of this study. Therefore, it is suggested that this study is followed up with the time series data and unbalanced panel data.

**Author Contributions:** All authors contributed equally to this work.

**Conflicts of Interest:** The authors declare no conflict of interest.

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
