# Peer review of "Regime-Switching Effect of Tourism Specialization on Economic Growth in Asia Pacific Countries"

_economies, doi:10.3390/economies5030023_

Round 1

Reviewer 1 Report

Although my overall assessment of the paper is positive, I would suggest the author(s) to do a minor revision of this paper before publishing.

Authors need to make additional effort in order to correct some deficiencies.

First of all paper needs to have a cleaner and better structure.

In the introduction, authors should clarify the importance of the topic, address main problems and describe the structure of the paper. In this case introduction is just focused to the literature background.

Furthermore, literature review section should include a short summary (few sentences) at the end of the section that should emphasize: 1. Are to date results conflicting, 2. What is missing in to date research, and 3. How exactly this paper will overcome the gaps identified in previous research and add value to theory / practice?
After reading literature review, I am missing the clear explanation what is new in this research compared to other to date studies; that is the core information that should be stated also in the abstract / introduction; that explanation is needed to motivate the reader to start reading the paper
In the section Literature review the most recent work quoted is from 2014 implies that recent publication from 2014 to 2017 have not  been considered.

Also I preferred the data sources are update at last from 1996 till 2014.

Methodology , have been fully elucidated.

Author Response

Q1:In the introduction, authors should clarify the importance of the topic, address main problems and describe the structure of the paper. In this case introduction is just focused to the literature background.

A1:The topic, address main problems and describe the structure of the paper has been marking the statements more clearly (line 66-69).

Q2:Furthermore, literature review section should include a short summary (few sentences) at the end of the section that should emphasize: 1. Are to date results conflicting, 2. What is missing in to date research, and 3. How exactly this paper will overcome the gaps identified in previous research and add value to theory / practice?

A2:In response to reviewer’s comments, these sections have been rewritten (line 171 to 176).

1.    According to the revision of literature review, we think the to date results are conflicting with using different methodology.

2.     From the past study, there are less research explore the hypothesis of different stages. Most of them aim to discuss the single stage tourism specialization.

3.  We use the nonlinear model to explore the relationship between them. By doing so, we can examine the hypothesis of two aspects which is tourism development led to economic growth or economic growth led to tourism development.

Q3:After reading literature review, I am missing the clear explanation what is new in this research compared to other to date studies; that is the core information that should be stated also in the abstract / introduction; that explanation is needed to motivate the reader to start reading the paper

A3: The core information of the paper has been marking the statements more clearly in abstract (line 4-7) and introduction (line 68-71).

Q4:In the section Literature review the most recent work quoted is from 2014 implies that recent publication from 2014 to 2017 have not  been considered.

A4:We made the literature review and references more current.  Moreover, we added statements about the recent work from 2014 to 2017 by reviewing more papers.

Q5:Also I preferred the data sources are update at last from 1996 till 2014.

A5:Thank you for the comment. In order to maintain the consistency and completeness of the data in this study. We used balance panel data over the period 1996-2009 as the research sample data. Will be the point we explore in the future.Moreover, we have revised the point on 5.3. Suggestions on follow-up research to improve the study in the future

Reviewer 2 Report

The paper is very interesting, but some comments could be done to improve it.

1).-Explain why the period of data analyzed finish in 2009, since then there are no more data to analyze? Why not to analyze the period to see the effect of the crisis.

2).-It would be interesting to explain which are the analyzed countries and which countries are in each of the regimes.

3).-It would be advisable provide an economic interpretation of the observed phenomenon.

Author Response

Q1: Explain why the period of data analyzed finish in 2009, since then there are no more data to analyze? Why not to analyze the period to see the effect of the crisis.

A1: 

1. Thank you for the comment. In order to maintain the consistency and completeness of the data in this study. We used balance panel data over the period 1996-2009 as the research sample data. Will be the point we explore in the future.Moreover, we have revised the point on 5.3. Suggestions on follow-up research to improve the study in the future

2.Thanks for the comment. This study aims to explore the influence of tourism development and economic growth by different stages of tourism specialization. We put emphasis on analyzing how the tourism development led to growth or growth led to development

Q2:It would be interesting to explain which are the analyzed countries and which countries are in each of the regimes.

A2:Thanks for the comment. This study aims to explore the relationship between tourism development and economic growth by the aspect of regional development. The different countries in each of the regimes will be the point we explore in the future.

Q3:It would be advisable provide an economic interpretation of the observed phenomenon.

A3:According to the comment of reviewer, this study revised the statement in the conclusion and the discussion by providing economic explanation.

Reviewer 3 Report

The literature review and references are not the most current. The most “fresh” reference is from 2013 and the most of them are even older. It is a pity as there are tons of studies, papers and books dealing with this topic. The same count for WTO (line 27), there are updates related with the world tourism on the yearly base.

I cannot agree with the statement: Since the Asia Pacific countries have had the rapidest tourism development over these past years, the international tourism receipts in this region are higher than those in any other places in the world (line 261 and 262). Although the change in the market share is the highest, international tourism receipts are higher in Europe.

For more current information see:

http://www.e-unwto.org/doi/pdf/10.18111/9789284418145

The quality of English language is not sufficient, paper needs serious proof reading by native speaker to avoid problems of word choice, grammar and sentence construction.

Author Response

Q1: The literature review and references are not the most current. The most “fresh” reference is from 2013 and the most of them are even older. It is a pity as there are tons of studies, papers and books dealing with this topic. The same count for WTO (line 27), there are updates related with the world tourism on the yearly base.

A1:We made the literature review and references more current by checking the website ( http://www.e-unwto.org/ ). Moreover, we added statement by reviewing more papers.

Q2:I cannot agree with the statement: Since the Asia Pacific countries have had the rapidest tourism development over these past years, the international tourism receipts in this region are higher than those in any other places in the world (line 261 and 262). Although the change in the market share is the highest, international tourism receipts are higher in Europe.

For more current information see:

http://www.e-unwto.org/doi/pdf/10.18111/9789284418145

A2: Thank you for the comment. We have revised the statement and the data. The revision is from line 281 to line 283.

Q3:The quality of English language is not sufficient, paper needs serious proof reading by native speaker to avoid problems of word choice, grammar and sentence construction.

A3: This paper has been review’s in some parts and over-all revised. Following the guidelines of the Economies and review’s instructions. The final draft will also be proved reading by a experienced scholar.
